# Scenarios for Sustainable Farming Systems for Macadamia Nuts and Mangos Using a Systems Dynamics Lens in the Vhembe District, Limpopo South Africa

Fenji Materechera *[ID] and Mary Scholes [ID]

School of Animal, Plant and Environmental Sciences (APES), University of the Witwatersrand, Johannesburg 2000, South Africa
* Correspondence: fenji.materechera@students.wits.ac.za

**Abstract:** Agriculture is arguably one of the most important economic sectors for South Africa's development as it is directly linked to food security. Farming systems in South Africa have been characterized by a duality where large-scale commercial farmers and small-scale farmers co-exist. The conventional approach to understanding agricultural production in the country has always viewed the two farming systems as mutually exclusive. The study argues that there are various points of interaction between the two kinds of farmers and by using a systems dynamics approach to evaluate the two farming systems this can be applied to agricultural decision making. Data were used to identify and characterise small- and large-scale farming systems of two tree crops (mangos—*Mangifera indica* L. and macadamia nuts—*Macadamia integrifolia* M&B.) in the Vhembe district of Limpopo South Africa. The interactions between the two different farmers are illustrated using Causal Loop Diagrams (CLDs) of the two farming systems under similar commodities. Results, presented as four conceptual scenarios, show that there are multiple points of interaction, such as the interdependence of farmers of macadamia nuts to meet export demands. Policy recommendations to strengthen collaboration between small-scale mango farmers and implement irrigation expansion for farmers who depend on rain-fed farming are discussed and present opportunities for the co-functioning of the two farming systems.

**Keywords:** interactions; scenarios; systems dynamics; agriculture; South Arica

## 1. Introduction

Agricultural productivity plays a crucial role in South Africa's food system and sustaining the country's food security. Farming systems in South Africa are characterized by a dichotomy where large-scale commercial farmers and small-scale farmers co-exist. This is part of the legacy of the apartheid system which relegated small-scale farmers to small portions of poor-quality land in what are known as the former homeland areas or Bantustans. The result of this has been the parallel functioning of these two kinds of farmers within the context of continuous change. Large-scale commercial agriculture is regarded as the main driver of national food security in South Africa [1]. In contrast to this, economically, small-scale agriculture in South Africa enhances local economic development as it is a source of employment and keeps most of the income local as the market is predominantly localised [2]. Hendriks [3] suggests that small-scale agriculture contributes to food security at a household level as socially, especially on traditional lands, the produce is first meant to feed the household. The two farming systems are therefore indispensable. According to Dixon et al., (2001) [4] farming systems are defined as " … *a population of individual farm systems that have broadly similar resource bases, enterprise patterns, household livelihoods and constraints, and for which similar development strategies and interventions would be appropriate. Depending on the scale of the analysis, a farming system can encompass a few dozen or many millions of households.*"

High value horticultural crops are becoming increasingly significant to the South African agricultural economy as there is a demand for them on the global market [5]. Some of the most popular high value crops grown in the country that are in high demand are avocados, mangos, litchis, pecan nuts, papayas, bananas and macadamia nuts. A number of these are cultivated in the Limpopo province of South Africa due to a favourable subtropical climate. Both large- and small-scale farmers are engaged in farming high value crops in the province intended for both export and supply to local markets. Land, though not the only driver, is a key driver of agricultural production in South Africa [6]. There are numerous factors pertaining to land that impact farmers' ability to successfully produce and contribute to the country's food system which include land tenure and its associated rights, soil quality, vegetation, topography, rainfall variability and water availability amongst various others. Government policy interventions and cross-sectoral initiatives have been targeted towards addressing these land factors with increasing focus on how small-scale farmers are affected by them. Examples of government policy initiatives geared at addressing land as a driver of production include the Upgrading of Land Tenure Rights Act, Act No. 112 of 1991 and the Restitution of Land Rights act, Act No. 22 of 1994 which was later amended in 2014 [7].

The common understanding of the context of South African farming systems is that the two farmers operate farming systems that are mutually exclusive. The current study challenges this notion by suggesting that the two kinds of farmers do interact on various levels, and this can be seen in the case of agriculture in the Vhembe district of Limpopo, South Africa. According to Labadarios et al., (2011) [8] one of the characteristics of farming systems is that they are able to produce the same outcomes in different ways provided they are exposed to similar conditions. It can be inferred that the variables and processes that comprise these systems will be different based on the scale at which the farmers operate therefore necessitating management practices that are specific to the farming systems. However, farming systems that exist in the same geographical location and may be exposed to the same vagaries such as extreme weather events due to climate change in South Africa, may experience overlap in the processes and management practices that are employed.

By using a systems lens to view farming systems in the Vhembe district of Limpopo, it is possible to conceptualize the future of South African agriculture and its contribution to food security by considering the plausibility of coupling the two kinds of farming systems. This is the conceptual basis of this study. The study aims to highlight the connectivity between the two main farming systems in South Africa using systems analysis as a tool for understanding. To this end, this paper will address two objectives, namely to identify the interactions between the two farming systems using Causal Loop Diagrams (CLDs) and to develop conceptual scenarios for the co-functioning of the two farming systems under similar commodities farmed in the Vhembe District of Limpopo South Africa. In understanding the nature of the interactions within and between the two farming systems it is possible to determine the feasibility of the two systems being coupled to achieve the goal of jointly meeting the country's food security needs at all levels. Scenarios can inform future decision making, research and policy recommendations.

## 2. Materials and Methods

### 2.1. The Study Area

Limpopo is one of the largest crop producing areas in South Africa and can be regarded as a key agricultural hub. The study took place in the Vhembe district which is a district municipality located in the Northern most region of the Limpopo province of South Africa (Figure 1).

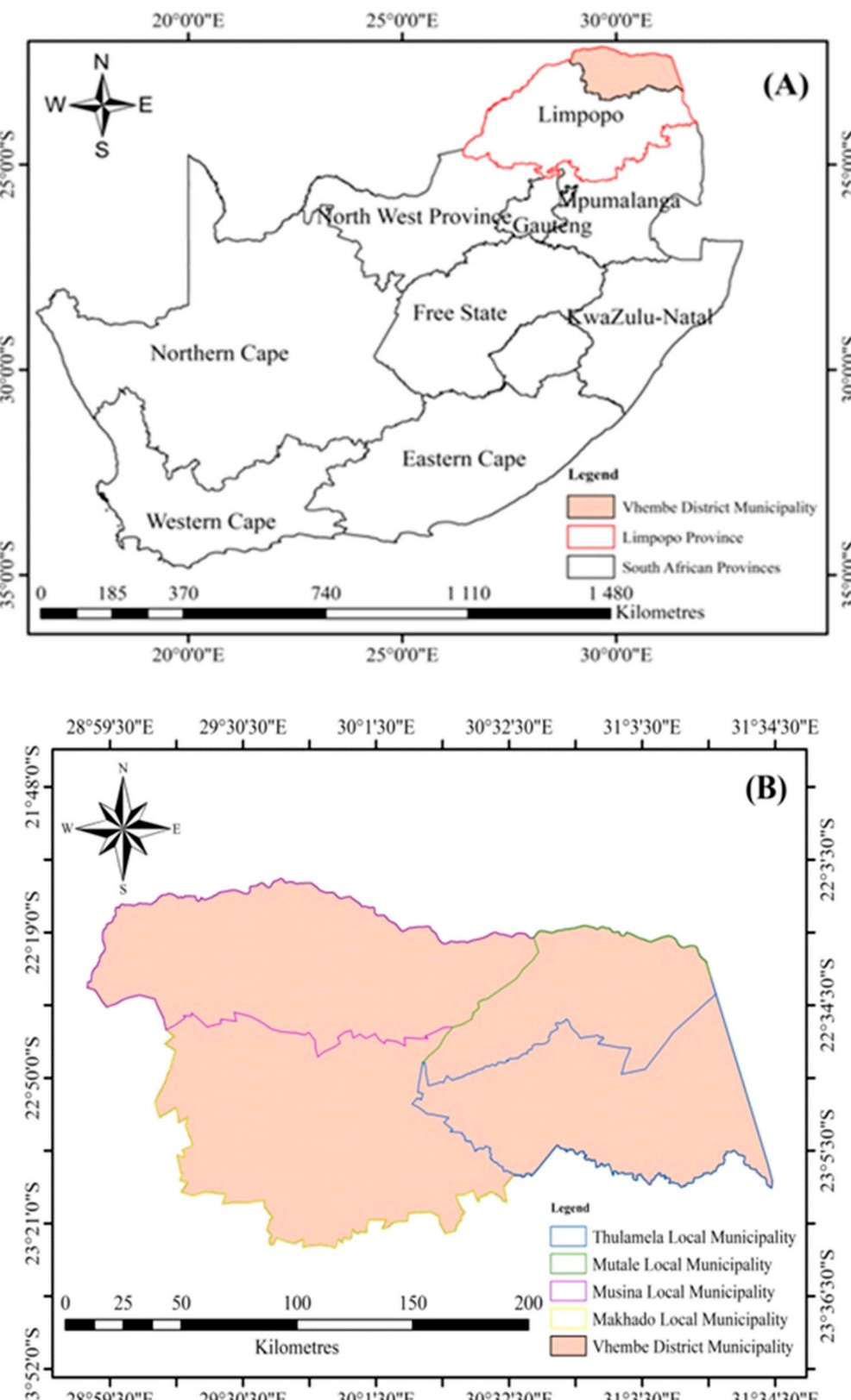

**Figure 1.** Map showing the location of (**A**) The Republic of South Africa' provinces and provincial boundaries, highlighting the location of the Limpopo province and the Vhembe district within the Limpopo province of South Africa and (**B**) shows the location of the four local municipalities within the Vhembe district.

The Vhembe district borders with Zimbabwe and Botswana to the north–east and Mozambique to the south–east passing through portions of the Kruger National Park [9]. The Limpopo province is comprised of five district municipalities of which the Vhembe district is one. The Vhembe district is further sub-divided into four local municipalities namely: Musina, Makhado, Mutale (renamed Collins Chabane) and Thulamela. According to the South African governance structure local municipalities are constituted by towns and their surrounding rural areas [10].

The Vhembe district has a land area of 2,140,708 ha of which only 247,757 ha is arable [11]. Agriculture is central to the livelihoods of the people in the Vhembe district and is a key contributor to employment. A reported 90% of rural communities located in the Vhembe district are dependent on agriculture to generate household income and sustain their livelihoods [12]. This is aligned with the geographical context of the Vhembe district where the district is located in an area that is predominantly rural [13]. Smallholder agriculture accounts for 70% of farming activities in the district while commercial agriculture accounts for the remaining 30% [14–16]. Numerous subtropical crops that contribute significantly to South Africa's agricultural economy particularly through exports are produced in the Vhembe district. Amongst these crops, are included commodities such as mangos, avocados, bananas, litchis, macadamia and pecan nuts. The census of commercial agriculture in 2017 recorded subtropical fruit and citrus as the biggest crop output in the district [17]. The Vhembe District Municipality's Local Economic Development Strategy in 2019 [18] reported that the Vhembe district produces 8.4% of the country's sub-tropical fruits and 6.3% of its citrus; overall amounting to 4.4% of South Africa's total agricultural output. Kom et al. [19] indicates that it is the southern side of the district i.e., the local municipalities of Thulamela and Makhado that is typically comprised of well-established white commercial horticulture farming. In contrast to this, the northern side is mostly semi-arid and is mainly utilized for livestock farming and game ranching. Horticulture in the northern region is very limited and restricted to areas where water is available.

In terms of water availability for agriculture, geographically the Vhembe district is located in a semi-arid area. Occasional droughts usually occur from May to August [13]. According to [20,21] small-scale farmers in the district mostly practice rainfed agriculture relying on seasonal rainfall which typically falls between November and March. Moeletsi et al., (2013) [20] documents that the average seasonal rainfall for the southern side of the district, identified earlier as the horticulture hub, ranges from 400 mm to 600 mm. With regard to soils, according to [14] the soils found in the southern region of the district vary significantly from one place to another; those with a higher clay and loam content tend to be found in the east and more sandy soils towards the west.

*2.2. Study Design*

Primary and secondary data were used to identify and characterise both small- and large-scale farming systems of three tree crops in the study area i.e., avocados, mangos and macadamia nuts. For the purpose of this paper avocados were excluded in the discussion as there was significant overlap between the interactions between large- and small-scale farmers of avocados and macadamia nuts in the study area. Mangos and macadamia nuts were selected as there were substantial differences in the interactions between the farmers of these tree crops which provided a useful means for comparison between commodities that enrich the discussion of the paper. Analysis was aimed at highlighting the connectivity of interactions within and between the two main farming systems with respect to the four drivers of production namely land, labour, capital and enterprise. Secondary data were derived from the official database of subtropical crops from the local Department of Agriculture, soil data and land type maps from the Agricultural Research Council (ARC), climate data from the Institute of Soil, Climate and Water (ISCW), related peer reviewed research papers and books. The target population was comprised of a combination of large-scale commercial and small-scale farmers of the three tree crops in the district. Initially, farms were selected based on data extracted from the subtropical database. Using a purposive

sampling method [21], the criteria for site selection were determined, namely commodity, farm size, gender of the farmer and farm location (village, town and municipality). This information was available for six subtropical commodities, namely bananas—*Musa paradisiaca* L. (23); litchis—*Litchi chinensis* S. (92); avocados—*Persea americana* M. (204); mangos—*Mangifera indica* L. (528); macadamia nuts—*Macadamia integrifolia* M&B. (184); and citrus—*Citrus sps* L. (90). According to the database there are a total of 1121 documented subtropical crop farmers in the Vhembe district. The database also showed that the three commodities selected in the study were the most commonly grown commodities in the district. Mangos were selected because they formed the largest number of farms documented in the database (528 farmers). Macadamias were selected based on their significance to the South African agricultural economy as high value export crops.

Thereafter, a systematic random sampling procedure was used to select farms to ensure equal representation of farm size. This was done because the study required both farmers with smallholdings and larger holdings. Initially three size categories based on the sizes that exist in the database were selected namely, small (1–5 ha), medium, (6–13 ha) and larger (14–20 ha and above). This was later narrowed to two categories i.e., small-scale (1–10 ha) and large-scale (11 ha and above) as these provided a continuum that was context specific to the study. The classification of small-scale farmers in the South African context is complex as size is not the only factor used to determine what constitutes a small-scale farm. Other factors such as enterprise, level of mechanization and technology employed, income from farming etc. are also taken into consideration [22]. This is further reflected in the use of numerous terms to describe these kinds of farmers such as subsistence, semi-commercial, emerging etc. [16]. For this reason, the researchers used their own criterion of size to classify small-scale farmers for the specific purpose of this study. In terms of location, farms were selected that reflected equal representation of the four local municipalities located within the Vhembe district namely Musina, Makhado, Thulamela and Mutale in order to provide a comprehensive overview of farming in the district. Lastly, with regard to the criterion of gender of the farmers, a random number generation method was used to ensure equal representation of the genders across all farms. This was achieved by allocating each farmer a number using the previously mentioned criteria and placing the written numbers in a container. Numbers were then randomly picked out by the researcher to add up to a total of 12 farms. Twelve farms were selected and were made up of four samples of each of the three tree crops across the four municipalities with two small-scale and two large-scale farms and an equal distribution of male and females. After completing the site selection, a more detailed characterisation of the two farming systems based on the three commodities in relation to the four factors of production followed. Primary data were obtained by way of in-depth, on-site interviews with individual farmers. Using a snowball sampling method [23] interviews were conducted with the aim of maintaining the originally selected sample size, The result of the snowball sampling technique that was employed produced the following samples: avocados (8), macadamia nuts (7) and mangos (4). In total, 19 farmers were selected for participation in the in-depth interviews based on their willingness to participate and availability. Due to numerous challenges in accessing farms based on their extremely rural locations, data were collected at only one point in time. This influenced the exceptionally small sample size which the authors acknowledge. For the purpose of this paper the sample refers to a total of 11 farmers (7 macadamia nut farmers and 4 mango farmers).

### 2.2.1. Data Collection

Face-to-face farmer interviews were conducted over the duration of the two visits to the Vhembe district between October and November 2020. Ethical clearance was obtained through the University of the Witwatersrand ethics committee (protocol number: H19/09/26). Clearance was also obtained from the local Department of Agriculture through an official letter of approval. A questionnaire was used as the main data collection instrument comprised of closed and open-ended questions with the aim of collecting qualitative

and quantitative data. Demographic information about the farmer was obtained through the questionnaire to obtain statistical data. Detailed information about various aspects of the four drivers of production in the context of the selected farm sites was also obtained. Open-ended questions were used to obtain more detailed responses from participants while close-ended questions were used to gather statistical information. The questionnaire was sub-divided into four key sections: land, labour, capital and enterprise.

Interviews were conducted by the researcher alongside a local who served as an interpreter due to language barriers. Interviews were mostly conducted in the local language of Vhenda. Key informant interviews were conducted with managers from processing plants for macadamia nuts and avocados. Study group meetings for the respective commodities were attended by the researcher in order to develop scenarios. These were information sharing sessions with various stakeholders (farmers, equipment suppliers, extension officers, researchers, grower association representatives, and government officials from the local Department of Agriculture) that allowed for interaction.

### 2.2.2. Causal Loop Diagram (CLD) Construction

The causal loop diagram (CLD) analytical tool used to represent the relationship between system variables and their dynamic feedback structures was constructed using Vensim modelling software (Ventana Systems Inc. 60 Jacob Gates Road Harvard, MA, 01451, USA, http://www.ventanasystems.com/, accessed on 4 September 2022) [24]. The overall structure of the CLD represents the links between large-scale and small-scale farmers of the two commodities (macadamia nuts and mangos) and the broader farming system. The CLDs hypothesize system behaviour and identify balancing and reinforcing feedbacks.

### 2.2.3. Development and Analysis of Scenarios

Scenarios were constructed using a scenario method known as 'deductive' [25] The deductive process uses multiple iterations of scenario drafts that are typically developed through stakeholder engagement facilitated through workshops [25]. Workshops allow for scenario deconstruction and revision which provide an opportunity to validate the plausibility of the scenarios. In the current study scenarios were created drawing from three iterations of scenario drafts based on (1) predominant themes on production issues arising within the farming systems of the selected commodities based on farmer interviews and secondary data (2) key informant interviews and (3) interactive study group meetings. Each narrative provided as much detail as possible with the aim of developing equally plausible futures based on a chosen time frame of 10 years (2022–2032). The 10-year time frame was preferred over a longer projected time as it provides a more realistic timeframe to imagine plausible futures based on current trends.

The process of creating scenarios involved three steps. In the first step key drivers and uncertainties surrounding production that emerged from interview responses were noted. Secondly, notes from conversations with key informants who were interviewed during field visits were used to give further detail to what these key drivers are which produced an outline of the narrative for each scenario. Key informants included technical managers for processing plants of macadamia nuts (Green Farms Nut Company and The Royal Macadamia) and representatives from the respective growers' associations (Macadamias South Africa i.e., SAMAC and the South African Avocado Growers Association i.e., SAAGA. Lastly, primary information was obtained from multistakeholder participation at local information sharing sessions termed study group meetings for the respective commodities which the researcher attended during the duration of the field work. These were used to further corroborate what the key drivers and uncertainties are and to produce the final two storylines outlining plausible scenarios for farming systems of the respective commodities in the Vhembe district by the year 2032. This iterative process is illustrated in Figure 2 below. The key driving forces of production were continuously narrowed with each iteration based on the most common themes recurring from feedback from the different participants. Themes were used as direction for what the key issues that the

scenario would address should be. A 2 × 2 quadrant of four key drivers based on a scale of uncertainty vs. impact ranging from low to high (Figure 2) was used to establish what the main subject of the scenarios would be. Issues for which farmers' responses reflected a high level of uncertainty were typically used as conversation points during information sharing sessions to probe what kind of solutions could be explored to address the challenge. These aided the writing of the scenario narratives. The interactions between farmers highlighted in the CLDs were used to evaluate the scenarios.

### 2.2.4. Data Analysis

Descriptive statistics were used for the analysis of quantitative data [26] by calculating percentages, averages and standard errors. Chi-squared and student t-tests [27] were used to compare the means across the two farm sizes and between the three commodities. Thematic analysis was used to analyse qualitative data [28]. Participants responses to open-ended questions concerning land variables relevant to the different commodities were transcribed. Thereafter recurring responses that were mentioned were identified as major themes. Based on the themes, percentages were calculated to classify them in order of importance. Predominant themes were triangulated with quantitative data from the questionnaire and secondary data to explain phenomenon.

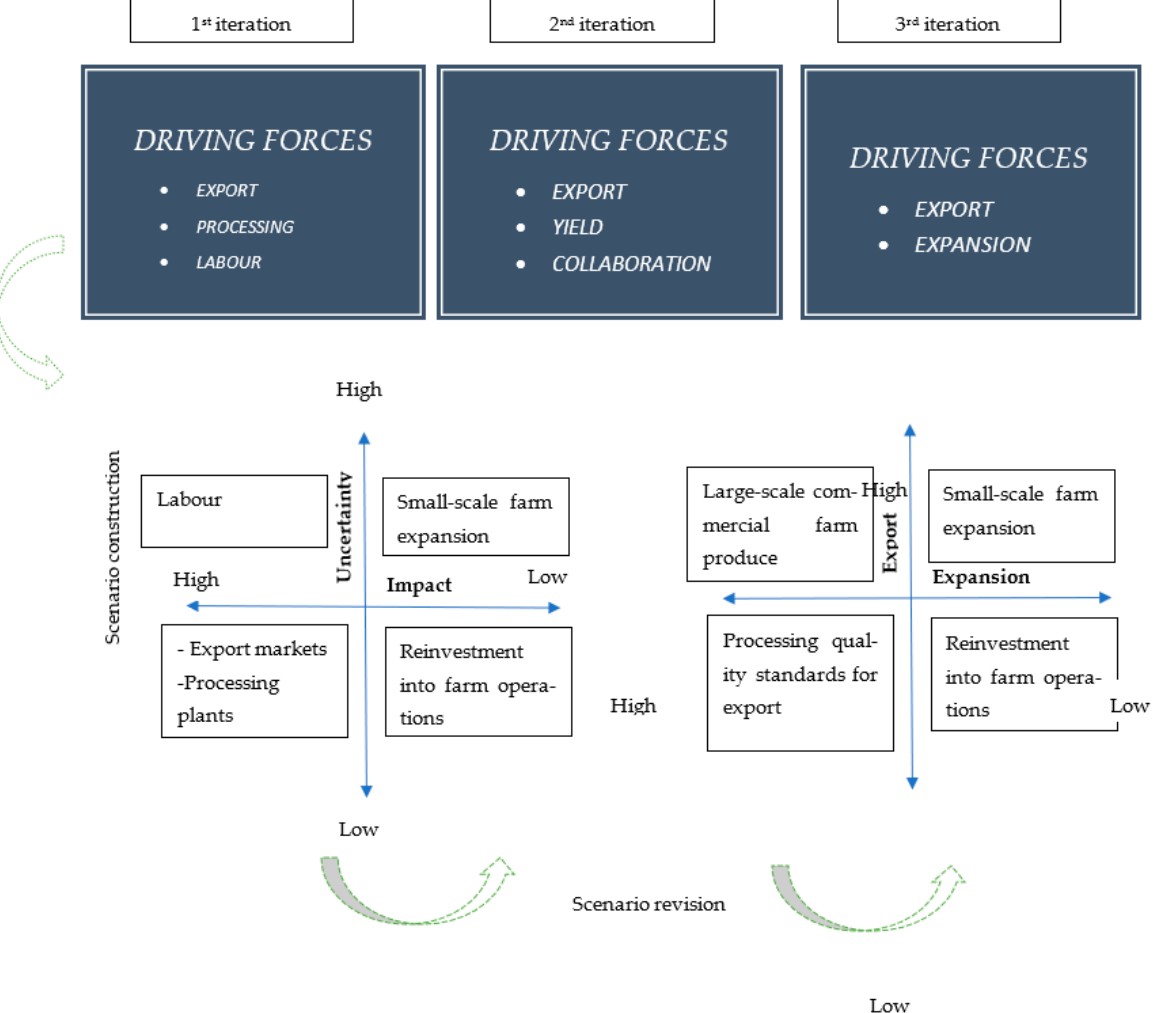

**Figure 2.** An example of the three iterations of the scenario development using the 2 × 2 quadrant of uncertainty vs. impact used for scenario construction for macadamia nuts based on the key drivers and uncertainties identified from interviews. Adapted with permission from Rameriz et al. [29] 2022. Rafael Ramirez.

*2.3. Conceptual Framework*

The two main farming systems that currently exist in South Africa can be found across the country. According to the FAO [30], both farming systems extend across the northern part of the country where the Limpopo province is located. The two farming systems are impacted by the same drivers of production i.e., land, labour, capital and enterprise [31], however respond to these drivers differently. The manner in which the two farming systems respond to the drivers of production may reveal the connectivity between the systems. A systems thinking approach is best suited to illustrate the connectivity between the farming systems and is therefore used for the study. Arnold and Wade [32] define systems thinking as "*a system of thinking about systems*". The same authors make the assertion that systems thinking must have three components in order to be defined namely elements i.e., characteristics, interconnections i.e., the way these interactions feed back into each other or relate, and a function or purpose. System dynamics (SD) is the understanding of the relationship between integrated systems elements and how they impact each other's behaviour [33]. The integration of systems elements is done by the incorporation of concepts such as stocks, flows, feedbacks, and delays, enabling the analysis of the dynamic behaviour of the system elements over time [34]. The approach is used to describe, model, simulate, and analyse complex systems with multiple interacting elements in terms of processes, information, organisational boundaries, and strategies [35]. This conceptual understanding of systems thinking, and systems dynamics is applied in the current study as a means by which to understand the systems being analysed. The farming systems in South Africa operate against the backdrop of constantly changing economic, political, environmental and socio-economic conditions. This is the context in which the current research is positioned. Although farming systems research and farming systems analysis are well established research fields [36], little attention has been paid to how farming systems will respond to change in the future with respect to drivers of production. The study seeks to provide foresight into future farming systems in a developing country with constantly changing parameters. According to [37] when scenario analysis is used in environmental change research an important objective is exploration. Scenarios can potentially assist users to consider surprising discontinuities and developments. Scenarios are defined as "*a set of conceptual systems of equally plausible future contexts often presented as narrative descriptions typically for the purpose of providing inputs for future work*" [29]. By using scenarios derived from a systems thinking viewpoint as a tool, the study identifies four scenarios for production, for two different commodities, in farming systems in the Vhembe district of Limpopo South Africa.

## 3. Results and Discussion

Results are presented here in three sub-sections. Firstly, a general (for South Africa and the Vhembe district) and more detailed (by yield and income) characterization of macadamia nut and mango farming systems is presented. Secondly, CLDs are presented, and the reinforcing and balancing feedback loops are described to improve our understanding of the interconnected variables impacting production of the respective commodities in the Vhembe district. Lastly, scenario narratives derived from the key factors highlighted by the CLDs and the iterative process of scenario development are presented.

*3.1. Characterization of Macadamia Nut and Mango Farming Systems*

Results revealed that by 2019 South Africa was the largest macadamia producer in the world with 19,500 ha under cultivation, producing over 50,000 tonnes per year. Over 95% of South Africa's macadamia nut production is exported annually [38]. According to [39] the average yield for macadamia nuts in South Africa was 1.43 tonnes per hectare in 2019. Only 7% of macadamia nuts grown in the country are consumed by the local market. The Limpopo province is the second largest macadamia nut production area in the county after the Mpumalanga province, and the Vhembe district ranks third in order of the highest macadamia nut contributing districts in the province [12].

Results showed that mango production in South Africa has been unstable in recent years. In 2019 a total volume of 68,633 tonnes of mangos was produced in the country during that production season [40]. This may be attributed to unfavourable weather conditions. The industry makes an important contribution to direct employment in mango production and processing. In terms of the market structure, the annual crop is either sold fresh through the national fresh produce markets and as exports or processed into atchar, juice or dried mangos. The majority of mangos exported from the Limpopo province are mainly from the Mopani and Vhembe district municipalities respectively. The total export value reported by the Limpopo province was R62 million in 2019 of which R3 million was reportedly from the Vhembe district [40]. Table 1 is a summary of the characterization of the two sets of farms based on selected criteria from the farms selected in the study.

**Table 1.** Characterization of farm size, farm type, tonnage, yield and income by commodity for one year (2019).

| Commodity | Farm Size (ha) | Farm Type | | Tonnage (t) | Yield (t/ha) | Gross Annual Income (ZAR) |
| | | Small-Scale | Large-Scale | | | |
|---|---|---|---|---|---|---|
| * Macadamia | 4 | ✓ | | 0 | 0 | 0 |
| Macadamia | 5 | ✓ | | 2 | 0.4 | 10,000 |
| Macadamia | 5 | ✓ | | 2 | 0.4 | 150,000 |
| Macadamia | 6 | ✓ | | 4.2 | 0.7 | 200,000 |
| **Mean ± SD** | 5 ± 0.8 | | | 2.7 ± 1.3 | 0.5 ± 0.2 | 120,000 |
| Macadamia | 34 | | ✓ | 17 | 0.5 | 300,000 |
| Macadamia | 93 | | ✓ | 47 | 0.5 | 35,000,000 |
| Macadamia | 1600 | | ✓ | 806 | 0.5 | 40,000,000 |
| **Mean ± SD** | 575.7 ± 887.6 | | | 290 ± 447.1 | 0.5 ± 0 | 25,100,000 |
| Mango | 2 | ✓ | | 3 | 1.5 | 12,000 |
| Mango | 2 | ✓ | | 3 | 1.5 | 10,000 |
| Mango | 10 | ✓ | | 4 | 0.4 | 150,000 |
| **Mean ± SD** | 4.7 ± 4.6 | | | 3.3 ± 0.6 | 1.1 ± 0.6 | 57,333 |
| Mango | 15 | | ✓ | 4.5 | 0.3 | 20,000 |

* The first farmer appearing on the table was a first-time farmer who had planted trees 2 months prior to the interview and therefore did not have any yield to record.

The average gross annual income from farming amongst participants ranged between R10,000 and R40 million between the two commodities. Results revealed that macadamia farmers obtained the highest farming incomes, in both large-scale farms, average of R25,100,000, and small-scale, average of R120,000 compared to mango, R20,000 for the large-scale farmer and an average of R57,333 amongst small-scale farmers, farmers. Results of the Pearson Correlations analyses show that there is a positive statistically significant correlation between average gross annual income and farm size amongst macadamia farmers (r = 0.763, $p < 0.01$), and a positive significant correlation between average gross annual income and farm size amongst mango farmers (r = 0.346, $p < 0.01$).

Results showed that 79% of participants were male while 21% were female. The general gender profile of participants skewed towards male participants in both farm sizes and across the two commodities with only 25% of female participants who were mango farmers and no female macadamia farmers. This gender distribution is characteristic of the patriarchal context of the Limpopo province as presented in other studies conducted in the Vhembe district. This distribution is a reflection of the cultural norms and values of the Vhenda people who predominantly reside there where men generally tend to be the owners of the land. This can be viewed as a constraint, as the demographics of the broader province of Limpopo indicate that most small-scale farmers in the province are women as a result of adult males being involved in migrant labour. The small sample size obtained in the study limits detailed analysis of this aspect. The CLD for the Macadamia nut farming systems is presented in Figure 3. Followed by an explanation of the feedback loops that were identified as part of the analysis.

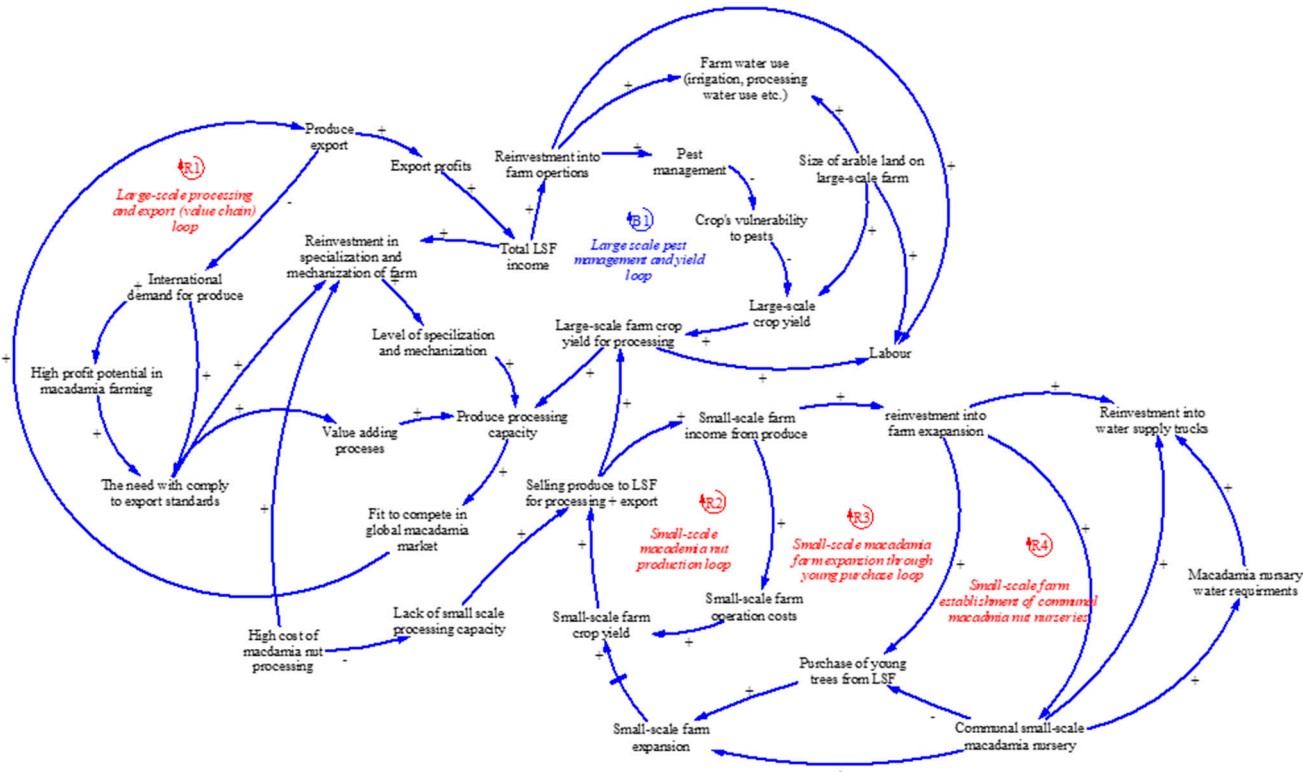

**Figure 3.** Causal loop diagram (CLD) showing macadamia nut farming systems in the Vhembe district, Limpopo. Arrows connect two or more variables of interest and are causal links that run in the stated direction. '+' = a positive relationship, indicating that the causality runs in the same direction (i.e., an increase in variable A will cause an increase in variable B and vice versa); '−' = a negative relationship, indicating that the causality runs in the opposite direction (i.e., an increase in variable A will cause a decrease in variable B and vice versa). The balancing feedback loops are numbered Bn and labelled in blue font. The reinforcing feedback loops are numbered Rn and labelled in red font. LSF refers to Large-scale farm/farming. Adapted with permission from Selebalo et al. [41] 2022. Itumeleng Selebalo.

### 3.2. Macadamia Nut Farming Systems

#### 3.2.1. R1 Large-Scale Processing and Export Loop

Macadamia nuts are the fastest growing tree crop industry in the country and their production is lucrative. The demand for macadamia nuts globally is high and South Africa is currently the largest producer (in tonnes per hectare) in the world [39,42,43]. Large-scale macadamia farmers in the Vhembe district produce macadamia nuts for export and are also owners or partners in processing plants such as the Royal Macadamia located in Thohoyandou, Limpopo and Green Farms Nut Company in Levubu, Limpopo. Some profits from export sales are reinvested into farm operations of which pest management forms a component. The most common pest control strategy used by large-scale farmers is integrated pest management (IPM). Large-scale farmers contract experts to monitor their fields and thereafter recommend management interventions. This IPM approach combines techniques such as the use of resistant varieties, biological control and habitat manipulation etc., to effectively tackle pest problems. Crop vulnerability to pests such as stink bugs is decreased through investments into pest management which resultantly impacts the total annual yield positively. There is a positive causal link between the large-scale farm yields and the capacity to process the nuts for export. Large-scale farmers are able to meet processing quality standards, therefore making them fit to compete in global export markets and to make profit from export sales, thus reinforcing a cycle of export market participation.

### 3.2.2. R2 Small-Scale Macadamia Nut Production Loop

Small-scale macadamia farmers in the Vhembe district contribute to the macadamia nut value chain in the province reinforcing the interdependence of the two types of farmers. Nuts, produced by small-scale farmers, are transported and processed at plants owned by large-scale farmers as small-scale farms do not have the required equipment for processing and export requirements (indicated by the negative relationship "−" that is shown in the arrow between the variables high cost of macadamia nut processing and lack of small-scale processing capacity). Income made from nut sales is used to finance farm operational costs.

### 3.2.3. R3 Small-Scale Macadamia Farm Expansion through Young Tree Purchase Loop

Small-scale macadamia farmers in the Vhembe district use some of the profits from nut sales to reinvest in the expansion of their farms by purchasing young macadamia trees from large-scale commercial nurseries in the province (these are found in Tzaneen and are sold at a cost of R60/tree). The expansion of small-scale macadamia farms will positively impact the yield over time as trees mature; this is indicated by the delay in the CLD (the short blue line across the positive arrow between small-scale farm expansion and small-scale farm crop yield) as the causal link between farm expansion and yield is not immediate. Macadamias are long-term crops taking on average four to five years from planting before cropping commences and six to seven years before commercially viable yields are produced. This reinvestment of profits reinforces a loop of continuous farm expansion.

### 3.2.4. R4 Small-Scale Macadamia Farm Establishment of Communal Macadamia Nut Nurseries

Some small-scale farmers have opted to establish their own nurseries through planting trees from the yield of previous harvests and grafting. Small-scale farmers then sell young trees to other small-scale farmers within close proximity eliminating the transport costs to nurseries further afield. The establishment of macadamia nut nurseries fosters interaction and interdependence between small-scale farmers and promotes growth in the small-scale macadamia enterprise thus reinforcing a loop of continuous expansion.

### 3.2.5. B1 Large-Scale Pest Management and Yield Loop

Large-scale commercial macadamia nut farmers in the Vhembe district are able to reinvest income from export profits into pest management to control prevalent pests and diseases. The more farmers able to invest in integrated pest management programs, the less vulnerable the orchards become to invasion by pests. As pest vulnerability is continually reduced through pest management, yield is increased through larger numbers of trees able to produce nuts in a balancing loop in the CLD (Figure 4).

### *3.3. Mango Farming Systems*
### 3.3.1. R1 Small-Scale Mango Farm Value Chain Loop

Small-scale mango farmers in the Vhembe district are the main suppliers of the local mango atchar (a pickled mango paste that is commonly eaten in the province forming part of a typically South African diet and sold at supermarkets) processing companies. Mango atchar processing factories (such as Gratchar located in Letaba, Mango Magic Atchar in Tzaneen and Levubu Atchar Veraardigers in Levubu) are located within the district therefore more easily accessible to the farmers. Farmers are able to make a profit from mango sales to atchar processing factories which are later reinvested into farm operations. An increased investment in farm operations results in better annual yields which ensures continued supply to processing companies creating a reinforcing loop.

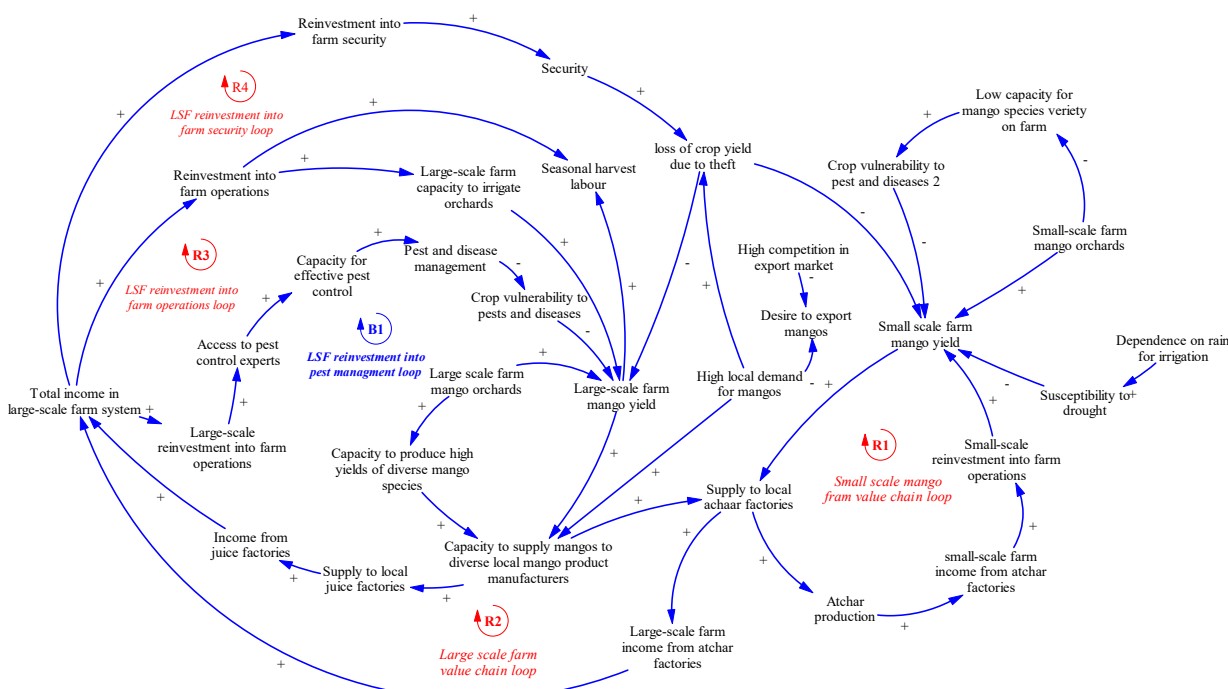

**Figure 4.** Causal loop diagram (CLD) showing mango farming systems in the Vhembe district, Limpopo. Arrows connect two or more variables of interest and are causal links that run in the stated direction. '+' = a positive relationship, indicating that the causality runs in the same direction (i.e., an increase in variable A will cause an increase in variable B and vice versa); '−' = a negative relationship, indicating that the causality runs in the opposite direction (i.e., an increase in variable A will cause a decrease in variable B and vice versa). The balancing feedback loops are numbered Bn and labelled in blue font. The reinforcing feedback loops are numbered Rn and labelled in red font. LSF refers to Large-scale farm/farming. Adapted with permission from Selebalo et al. [41] 2022. Itumeleng Selebalo.

### 3.3.2. R2 Large-Scale Mango Farm Value Chain

Large-scale mango farmers in the Vhembe district produce larger annual yields compared to small-scale farmers; these are comprised of more than one variety of mango species therefore enabling them to supply mangos to diverse markets i.e., juice manufacturers, dried fruit and mango atchar processing factories within the district, fresh produce markets, informal markets and supermarkets in other provinces. None of the farmers interviewed indicated that they supply mangos for export. The total income made from the sales to these diverse markets is used to reinvest in farm operations of which irrigation forms a part., only the large-scale farmers indicated that they irrigate while all small-scale farmers stated that they rely on rainfed agriculture. With an increased capacity to irrigate there is an increase in yield which allows farmers to supply the diverse markets creating a reinforcing loop.

### 3.3.3. R3 Large-Scale Mango Farm Reinvestment into Farm Operations Loop

Large-scale mango farmers in the Vhembe district are able to reinvest profits from sales into farm operations which include labour. Large-scale farmers are able to reinvest in paying seasonal labour during harvest time unlike small-scale mango farmers who rely on family members to harvest mangos.

### 3.3.4. R4 Large-Scale Farmer Reinvestment into Farm Security Loop

Large-scale mango farmers are able to reinvest the profits from selling produce into improving security on the farm. Theft is an ongoing challenge to the mango farmers as mangos can be sold locally by vendors in the district. The lack of adequate fencing means that surrounding communities can easily access the trees and steal large quantities of

mangos (one farmer reported "*last year I was only able to harvest about a quarter of my whole farm, the rest was stolen. All that work for nothing.*") significantly impacting the quantities of mangos available for sale to markets. When farmers increase the investment in security i.e., fencing, patrol guards and watch dogs this decreases the loss of the crop due to theft and increases the overall annual yield creating a reinforcing loop.

### 3.3.5. B1 Large-Scale Farmer Reinvestment into Pest Management

One of the areas in which large-scale mango farmers in the Vhembe district reinvest their profits from sales is in pest management. Farmers are able to outsource pest control experts to inform their pest management activities, therefore increasing their capacity for effective pest management by implementing an integrated pest management approach that is capital intensive. Results showed that farmers made use of both spraying of pesticides and herbicides to this end. Continuous investment into effective integrated pest management decreases the vulnerability of orchards to pest invasion which creates a balancing loop that ensures good annual yields enabling continued supply of mangos to the diverse markets that large-scale farmers have access to. Due to the plethora of resource constraints experienced by small-scale farmers in the region investment in IMP is generally limited. Large-scale farmers are better equipped to make this investment.

### *3.4. Scenarios*

### 3.4.1. The Macadamia Gold Rush

The global demand for macadamias continues to increase as there is an increasing public knowledge of the numerous health benefits of tree nuts and nut oils. This demand has been the key factor in market expansion. According to [44], the global macadamia nut market is expected to grow at a compound annual growth rate of 10.7% from 2021 to 2028 to reach USD 2.95 billion by 2028. South Africa remains one of the largest producers in the world and this can influence future production trends as farmers in the country aim to align with global market demands. Small-scale farmers' heavy reliance on large-scale farmers for processing in order to participate in global market supply will continue if there are no opportunities created for them to compete in terms of processing capacity. A wide range of role players need to be involved within the macadamia nut industry in order to make it competitive, efficient and dynamic. Small-scale farmers can only expand the industry if they have access to land and tenure security; results revealed that higher proportions of small-scale farmers (71%) farmed on communal land compared to large-scale farmers (29%). This speaks to the on-going land tenure reform dialogues in South Africa and the need to urgently address tenure rights of small-scale farmers in the country. In order to sustain large-scale and small-scale macadamia farmer interdependence in a manner that is mutually beneficial and equally beneficial to the country's agricultural economy, small-scale farmers need to be incorporated into the value chain in a more prominent way.

### 3.4.2. Exploring the Possibilities of Strengthening Small-Scale Farmer Collaboration

There is an increase in interest to farm macadamia nuts amongst small-scale farmers as the monetary gains become increasingly evident. This is well depicted in participant's contribution at a study group meeting; "*everyone is going into macadamias now because that's where the money is. If I could, I would convert my whole farm into only macadamias*". Despite this growing interest, small-scale farmers would not be able to cope well with a major ecological or market failure in macadamia nut farming if they relied solely on the single crop. Intercropping is a highly beneficial practice for small-scale farmers as it has been established in literature on farming in South Africa that small-scale farming plays a dual role of being a source of household food security as well as generating income from sale of surplus [3]. For this reason, the practice of a monoculture cropping system, typically characteristic of large-scale commercial farmers in the country is not ideal for the small-scale farmer sustainability. The need to diversify their farming and develop a system where they are able to benefit from the yields of other food crops and supplement household income

from farming with the profits from the sales of high value crops is essential. This approach should be encouraged to maintain small-scale farmers' significance as contributors to household food security as indicated in the introduction of the paper. The expansion of small-scale macadamia nut farming should therefore be supplementary to existing farming practices.

The move towards expansion of macadamia nut farming is seen in the establishment of nurseries amongst small-scale farmers from the yields of previous harvests. This is an attempt at breaking away from their dependence on commercial tree suppliers and creating a level of independence. If successful, this initiative has the potential to grow small-scale farmer's producing capacity over time. Establishing their own nurseries also presents a premise for small-scale farmer collaboration that may yield better production results. If small-scale farmers came together to increase their yields, they can continue to supply large-scale commercial processors and enter the export market at a more competitive level. The benefits of the outcomes of this interaction and interdependency between the two kinds of macadamia famers are not balanced. Small-scale farmers may not obtain profit to the full value of their produce as they only provide raw produce and are paid accordingly. Their large-scale farmer counterparts on the other hand obtain a higher profit as the produce that is sold to export markets is now value added after processing. Based on this imbalance, there is a need to explore more innovative approaches to collaboration between small-scale farmers for the purpose of enabling them to process macadamia nuts independently. Small-scale farmers could potentially band together to either rent or co-own processing facilities that they would collectively use instead of solely relying on large-scale commercial farmers for processing. There is potential to develop equipment more suited and more affordable for small-scale farmers if this is made a research and policy directive. This should serve as a model to inform government support for capacity building amongst small-scale farmers; with the aim to enable them to increase their profits from growing macadamias so that there is a balance in the benefits derived from growing macadamia nuts for both large- and small-scale growers. Lastly, although the main focus of macadamia supply in recent years has been international markets, there is potential for macadamias to become a highly sought-after commodity in local markets with changes in the South African food system leaning towards a more healthier food focus. Both large- and small-scale farmers can work together to explore how to optimize opportunities and risks.

### 3.4.3. Mango Supply Driven by the Demand of the Market

Mangos are highly perishable therefore necessitating careful control of packaging, transportation and distribution. This influences the South African mango value chain significantly. Unlike macadamia nuts, the market demand for mangos from farmers in the Vhembe district appears to be more localized than international. The ability to grow different cultivars based on favourable climatic conditions enables farmers in the district (both large- and small-scale) to target specific markets based on the type of mangos they produce e.g., juice making factories, processing factories (for dried fruit and mango atchar), local and provincial fresh produce markets. Currently, large-scale mango farmers from the Vhembe district supply produce to juice processing factories while small-scale famers supply atchar processing factories. This is mainly an issue of accessibility, as most atchar processing companies are located within the district much closer to where the small-scale farms are situated. They are therefore able to transport the produce to these factories faster. This becomes a more economically viable option for small-scale farmers as mangos rot easily and therefore may result in losses when they attempt to transport to further distances where the juice processing factories are found (in some cases outside of the district and province where they live). Agro-processing is the single largest market for mangos in South Africa [40]. According to the database of local subtropical fruit farmers, mango farmers make up the largest number of farmers in the district presenting an opportunity for economic gain; this however does not align with the success of the mango market distribution when compared to that of macadamia nuts and avocados. The economic

profitability of processing mangos into juice is high as value is added to the raw produce once it is in the form of juice and can be preserved for longer than the mangos in their natural state. Atchar is also highly profitable as it is a popular choice as part of a low to medium income South African diet. It has a long shelf life due to the manner in which it is preserved, therefore presenting a viable economic investment. There is also potential for atchar to be sold as an export product to other countries in the region and abroad. Exploring possibilities of collaboration between farmers and agro-processors can possibly expand the value chain for the benefit of all stakeholder. Given the heterogeneity of the local mango demand, farmers in the district can invest in a more targeted approach to growing mangos, focussing on the niche markets. Mango cultivars that ripen earlier in the season are more favourable for atchar processing as opposed to cultivars that ripen mid to late season which are more suitable for the juice market, however the risks associated with ripe fruit are high e.g., theft, flies, pests etc. Solutions need to be found to minimize theft and may be associated with price control.

### 3.4.4. Focus on Irrigation

One of the greatest challenges for both large- and small-scale mango farmers is their reliance on rainfed agriculture as the area is semi-arid and prone to droughts. Interview responses revealed that reliance of rainfall for irrigation was the sole source of water for irrigation for mango farmers with mature orchards; localized groundwater accessed through drilled boreholes was not indicated. One of the numerous impacts of climate change is that rainfall patterns are shifting, therefore sole reliance on rainfall for cultivation is not beneficial. This is a constraint that is already recognized and is an ongoing concern for mango farmers of different scales however, small-scale farmers are particularly vulnerable to this problem and therefore need be given more attention. Irrigation is a critical factor in farmers' success and capacity to supply markets. Systems thinking is a valuable tool for finding solutions where trade-offs are involved. The mango industry needs to collaborate with water management representatives in order to maximise on production.

### 4. General Discussion

The value of systems thinking is illustrated in the CLDs and the scenarios for the two commodities, macadamia nuts and mangos. In understanding the interconnections between variables and the degree to which they impact each other in the present, it becomes possible to adopt a more holistic approach to decision making that informs policy and action for the future. The current study provides evidence that suggests that the coupling of large- and small-scale farmers is a viable option for agricultural development in South Africa. If both farmers can equitably contribute to the country's agricultural economy albeit through different means, it is possible to envision an economy that is supplied by the joint operation of both kinds of producers. The success of this kind of approach hinges on the implementation of a multi-pronged intervention strategy which addresses related issues simultaneously. Examples of the need for this multi-pronged intervention strategy have been highlighted in the scenarios through: (1) the need to address small-scale macadamia farmer tenure security in order for small-scale farmers to successfully expand their industry and to collaborate with one another alongside prioritizing supplying export markets; (2) the need to explore the potential of alternative sources of irrigation amongst small-scale-farmers beside their reliance on rain-fed agriculture. Sufficient irrigation could potentially improve their yields which can resultantly increase income from sales to local markets over time. Increased income from farming can enable farmers to reinvest in long-term transportation solutions that assist in accessing diverse markets. This serves as a good example of the interconnectedness of variables in the systems.

The government model for land redistribution in South Africa over the past two decades has been centred around the need to address historical inequalities in land distribution that favoured a minority of large-scale commercial farmers over the majority of small-scale farmers who were predominantly black. This is a highly contentious and

politicized issue considering South Africa's history of an apartheid system. According to Materechera and Scholes, 2022 [6], the issue of overlapping use rights on communal land further complicates the challenge of a lack of tenure rights for small-scale farmers. Small-scale farmers have to contend with other community members who use the land for multiple other purposes e.g., firewood and grazing before they can consider participating in commercial activities. This presents itself a great constraint. Existing policy interventions surrounding land tenure security have mostly been targeted at land reform to improve the commercial status of previously disadvantaged farmers located in the former homeland areas. In order to be successful in this, policy directives should also include socio-economic strategies to address the issue of overlapping use rights in communal land. There is an urgent need for expansion as far as small-scale farmer irrigation systems are concerned. The potential for sustainable irrigation expansion thus becomes a factor that should inform research, decision making and policy development so that small-scale mango farmers can increase their yields and improve their market competitiveness. The application of innovation for more climate change friendly irrigation systems that are affordable and accessible to small-scale farmers becomes a necessity. The adoption of "soft-path" water harvesting for irrigation [45] is a plausible solution. This approach to bringing irrigation to rain-fed croplands involves capturing water resources in small and check dams as an alternative to the conventional centralized, capital-intensive irrigation projects that tend to be large.

Integrated pest management has the potential to be mutually beneficial for both farmers provided they are co-located. For example, a large-scale commercial farmer who implements IPM on their farm can have a positive knock-on effect on an adjacently located small-scale farmer of the same commodity. Small-scale farmers can use less capital-intensive, non-chemical approaches to IPM such as the cultivation of push and pull crops [46,47] in order to augment the activities of large-scale farmers that tend to be more capital-intensive. The implementation of this approach cancels the necessity of every area under cultivation with the same commodities to have a comprehensive IPM system in place. There may be areas without extensive IPM but benefit from being adjacent to farms that do. If farmers are willing to collaborate and make this trade-off, potentially with agreement on a certain kind of compensation for the benefits received, this interaction can foster future coupling of the two farming systems to achieve a common goal. This potential has not been explored in the current study, therefore there is no evidence of a willingness to collaborate on a regional IPM approach; this presents an opportunity for future research. Historically, the needs of the two kinds of farming systems have been addressed independently. This has contributed to the way in which they have continued to operate as separate entities. The study proposes that a systems thinking approach should inform decision making in the future.

## 5. Conclusions and Recommendations

Ultimately the question of whether coupling of the two farming systems in the context of meeting the country's food security needs at both national and household levels is a viable option is revisited. In understanding both the positive and negative implications of the interactions between the two groups of farmers, the scenarios derived in the study attempt to present evidence to support the conclusion of whether or not the two systems can ultimately work collaboratively in achieving food security at all levels in the future, as opposed to doing so independently as the current situation suggests. If decision making is informed by the application of systems analysis this may be an achievable goal. The study has shown that there is connectivity within and between large-scale commercial and small-scale farming systems in the Vhembe district. Applying systems analysis has shown that there are numerous points of collaboration across the two types of farmers. The use of systems analysis has also shown that the respective farmers who are co-located, respond to the drivers of production differently though farming the same commodity. This illustrates the potential for the coupling of the two farming systems and a transition from the historic dichotomous context of South African farming systems. There are research organizations

such as the regional systems analysis community of the International Institute for Applied Systems Analysis (IIASA) in South Africa which is active and can help foster a systems approach to coupling the two farming systems. Collaboration between such research entities and other stakeholders in the food system value chain across different spheres of government, agribusiness and the private and public sectors can produce favourable results.

Future studies on farming in South Africa should view farming as systems incorporating the whole value chain of any commodity. This should include the social, financial and environmental values and impacts. The conceptual scenarios developed in the study could be a basis for further evaluation to determine their feasibility under various predicted changes such as those presented by the present and foreseeable impacts of climate change. The use of scenarios is recommended as a tool to inform similar studies on farming systems in South Africa e.g., Trade-offs in the adoption of IPM. Future studies must collate data from a larger sample size.

**Author Contributions:** Conceptualization, F.M. and M.S.; methodology, F.M.; software, F.M.; validation, F.M.; formal analysis, F.M.; investigation, F.M.; resources, M.S.; data curation, F.M.; writing—original draft preparation, F.M.; writing—review and editing, F.M. and M.S.; visualization, F.M.; supervision, M.S.; project administration, M.S. and F.M.; funding acquisition, M.S. All authors have read and agreed to the published version of the manuscript.

**Funding:** This research was funded by the National Research Foundation and the Department of Science and Technology, through the funding of M.C.S's SARChI Chair in Global Change and Systems Analysis (Grant number 101057).

**Institutional Review Board Statement:** The study was approved by the Ethics Committee of The University of the Witwatersrand (H/19/09/26 on 22 October 2019).

**Informed Consent Statement:** Informed consent was obtained from all subjects involved in the study.

**Data Availability Statement:** Data supporting results reported in this paper namely the database of sub-tropical fruits from the Department of Agriculture in Thohoyandou, Limpopo, South Africa can be obtained from the authors upon reasonable request.

**Acknowledgments:** The farmers from the Vhembe District are thanked for their contributions and hospitality.

**Conflicts of Interest:** The authors declare no conflict of interest. The funders had no role in the design of the study; in the collection, analyses, or interpretation of data; in the writing of the manuscript, or in the decision to publish the results.

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
