# Peer review of "Scenarios for Sustainable Farming Systems for Macadamia Nuts and Mangos Using a Systems Dynamics Lens in the Vhembe District, Limpopo South Africa"

_agriculture, doi:10.3390/agriculture12101724_

Round 1

Reviewer 1 Report

1. The title of the manuscript may be revised to include macadamia and mango - the two were the two crops covered by the study.

2. May include the scientific names of the fruit plant the first tme they are mentioned.

3. Figure 1 - increase the font size to make the text clearly visible.

4. Section 2.2.1 - what information was obtained using the questionnaires?

5. Section 4 - may be re-named "Conclusions and recommendations; include recommendations of the study

Reviewer 2 Report

The presented manuscript is an original scientific work. The authors carried out a deep analysis of interactions between the two different farming systems from point of view of creating of sustainable farming systems in South Africa. The article will undoubtedly be of interest to the readers of the journal.

The only comment to the authors concerns the conclusion. It may be recommended to divide the section "4. General Discussion and Conclusions" into two separate sections.

Reviewer 3 Report

Agriculture (ISSN 2077-0472)

Manuscript ID  agriculture-1928475

Scenarios for future sustainable farming systems in South Africa using a systems dynamics lens, a case study of the Vhembe district, Limpopo South Africa

Authors

Fenji Materechera, Mary Scholes

I'm not so enthusiastic about this paper.  It aims to analyze large and small-scale producers largely of macadamia nuts in South Africa using a "systems dynamics approach."  Sounds promising.  They develop "causal loop" diagrams to define the dynamical systems for the macadamia and mango farming.  (The authors should check their spelling of "causal" – in their abstract they use "casual" which caught my eye!)

There are two things about this paper that disappoint.  First, it appears to be entirely qualitative – the cartoon causal diagrams are not used to do anything quantitative, either numerically or in terms of dynamical systems.  Second, the conclusions don't really seem very potent – I don't see much new "fruit" from the attempt to use dynamical systems thinking.  Some journalists have sharper analysis without the systems jargon. 

My review may seem harsh.  Perhaps I am missing something that other reviewers will discern.  Or perhaps the authors can produce an effective rejoinder.   I'm from a dynamical systems background myself, and maybe my perspective reflects that.   

Reviewer 4 Report

This is an excellent manuscript that takes a novel approach very well implemented.  One topic that could be included in the general overview of the regions farm systems is farmer gender.  The methods took great lengths to include male and female farmers in equal numbers but the authors did not indicate if both small and large farms are split roughly equally among male and female farms.  The interview numbers were small so it is understandable if analysis by gender is not possible, but in general are there constraints or resource access issues that are gender dependent? 

Other comments are annotated in the attached file. 

Round 2

Reviewer 3 Report

No further suggestions for authors.